# "Are Adversarial Phishing Webpages a Threat *in Reality*?" Understanding the Users' Perception of Adversarial Webpages

## ABSTRACT

Machine learning based phishing website detectors (ML-PWD) are a critical part of today's anti-phishing solutions in operation. Unfortunately, ML-PWD are prone to adversarial evasions, evidenced by both academic studies and analyses of real-world adversarial phishing webpages. However, existing works mostly focused on assessing adversarial phishing webpages against ML-PWD, while neglecting a crucial aspect: investigating whether they can deceive the actual target of phishing—the end users. In this paper, we fill this gap by conducting two user studies ($n$=470) to examine how human users perceive adversarial phishing webpages, spanning both synthetically crafted ones (which we create by evading a state-of-the-art ML-PWD) as well as real adversarial webpages (taken from the wild Web) that bypassed a production-grade ML-PWD. Our findings confirm that adversarial phishing is a threat to both users and ML-PWD, since most adversarial phishing webpages have comparable effectiveness on users w.r.t. unperturbed ones. However, not all adversarial perturbations are equally effective. For example, those with added typos are significantly more noticeable to users, who tend to overlook perturbations of higher visual magnitude (such as replacing the background). We also show that users' self-reported frequency of visiting a brand's website has a statistically negative correlation with their phishing detection accuracy, which is likely caused by overconfidence. We release our resources [1].

## 1 INTRODUCTION

After nearly three decades of research [30], phishing attacks are still rampant. According to the FBI's 2022 crime data [2], phishing is the topmost form of cybercrime, with reported victim loss allegedly increasing by over 1000% since 2018. In this context, phishing *websites* are a type of online scam used by attackers to steal sensitive information such as login credentials, financial information, or personal data. To increase their effectiveness, phishing websites aim to mimic legitimate ones [6], thereby tricking unaware and distracted victims—who may not notice subtle differences in their appearance.

Recently, numerous automatic Phishing Website Detectors (PWD) have been proposed, which can rely on blocklists [58], or be entirely data-driven [10]. The former works by checking whether a given website is included in their (public or private) blocklist, which consists of URLs (collected, e.g., from well-known repositories—such as PhishTank [3]). However, blocklist-based anti-phishing methods, despite their low false positive rates, cannot detect "novel" phishing websites [75]. These shortcomings can be compensated via data-driven techniques. Among these, Machine Learning (ML) algorithms seek to autonomously learn (by "training" on a given dataset) to identify patterns that may not be discernible to the human eye. The remarkable performance of ML methods in computer vision [47] led to many efforts to investigate their effectiveness in various fields—including that of phishing website detection. In particular, ML-based phishing website detectors (ML-PWD) can detect previously unseen phishing websites while maintaining low rates of false positives [10], which can be achieved by analyzing either textual or visual features from any given webpage (e.g., [17, 52]).

**Motivation.** Machine learning has now become mainstream even for the detection of phishing webpages [27]. However, ML is prone to evasion attacks, which entail crafting an "adversarial phishing website" (APW) by introducing imperceptible perturbations (located, e.g., in the HTML [10], or in some visual element [48] of a webpage) that fool an ML-PWD. Unfortunately, security practitioners persist in not addressing such a threat [9] (despite abundant alarms from academia [61, 64]). In this context, we observe that recent interview studies [19, 36, 54] about adversarial ML (AML) in practice are based on the participants' (self-reported) understanding of AML's concepts, thereby focusing on the question "What is the practitioners' awareness of AML?". We argue that to (i) establish whether AML is truly a threat and, if so, (ii) convince practitioners to take AML into consideration while designing their ML systems, the focus should be on the question "*What is the impact of AML on the end-users in practice? That is: does AML fool users as much as it fools ML models?*". This paper revolves around investigating this dilemma for phishing website detection. Compared to existing works that only focus on using AML to attack ML-PWD (e.g., [10, 48]), our work advances existing knowledge by examining how *human users* perceive adversarial phishing webpages that evade ML-PWD.

**Problem Statement.** To explore the users' perception of APW, our paper revolves around answering four research questions (RQ):

*RQ1* Are adversarially perturbed phishing webpages more easily detectable by users—w.r.t. unperturbed ones? (§5.2)

*RQ2* Are some perturbations more likely to deceive users? (§5.2)

*RQ3* How much do users' background (e.g., age, gender, expertise) correlates with their phishing detection skills? (§5.3)

*RQ4* What cues do users typically look for (and potentially rely on) to judge the legitimacy of any given website? (§6)

To answer our RQ, we conduct (§4) two user studies ($n$=470). The first focuses on assessing how well users can distinguish legitimate webpages from "unperturbed" phishing webpages. The second is to assess how well users can distinguish "adversarial" phishing webpages from legitimate webpages. Overall, we obtained over 7k responses encompassing various classes of webpages including: legitimate and 'unperturbed' phishing webpages, four types of APW (crafted through well-known AML techniques), as well as APW "from the wild Web" that bypassed production-grade ML-PWD (§3).

**Contributions.** After analysing the results of our user studies both quantitatively and qualitatively, we derive three key-findings.

(1) **Adversarial phishing is a threat to both users and ML.** In particular, three out of the four adversarial perturbations we considered have comparable effectiveness in deceiving users when compared to unperturbed phishing webpages—but the latter cannot bypass the ML-PWD. We argue that user studies

are a necessary step that is currently missing in most AML research on phishing detectors (see §2). Specifically, *it is crucial to compare adversarial phishing webpages with unperturbed phishing webpages* to make sure APW do not sacrifice effectiveness against users in favor of an improved evasion rate.

(2) **Not all adversarial perturbations are equally effective.** In particular, adversarial webpages with added typos are more noticeable to users, as confirmed by statistical tests. The reasoning provided by participants also indicates that textual indicators play a major role in their decision-making process. In addition, we verify that adversarial phishing pages "from the wild Web" (which bypassed production-grade ML-PWD) are *more detectable by users than unperturbed phishing pages*.

(3) As a surprising and counter-intuitive observation, **users' self-reported frequency of visiting a brand's website has a statistically significant *negative* correlation with their phishing detection accuracy**. Users who claimed to frequently visit websites of a given brand performed worse on the phishing webpages targeting this brand. We suspect this is correlated to prior findings that familiarity leads to overconfidence [62, 78]

Finally, our work can serve as a benchmark for future research on evasion attacks against ML-PWD, since it facilitates *assessing the effectiveness on end users* of the proposed attacks. To this purpose, we release our user study questionnaires, codebook, data, and code we developed [1]. We will also submit our tools for artifact evaluation.

## 2 BACKGROUND AND RELATED WORK

To set the stage for our contribution, we raise the attention on some simple security concepts, which we use as a scaffold to position our paper within existing literature. We provide exhaustive background (covering ML-PWD and adversarial ML) in Appendix C.

**Phishing in a Nutshell.** From a security standpoint, the goal of a phisher (i.e., the attacker) is to trick a *human user* to, e.g., input their private (or sensitive) data, or click on a malicious link.

> **REMARK:** bypassing a given detector (despite being necessary) is not sufficient for a phishing webpage to be successful.

Given the above, all those papers (e.g., [10, 11, 24, 48, 56]) showing that ML-PWD can be evaded via "adversarial perturbations" – while useful for investigating some robustness properties of ML – could hardly provide a compelling case that "adversarial examples are a problem *in reality*". Indeed, doing so would necessitate a double form of assessment, entailing both machine and human: first, it is necessary to craft an adversarial webpage and show that it bypasses a functional ML-PWD (i.e., a false negative); then, it is necessary to assess whether humans (i.e., the true target of phishing) are still tricked by such a webpage. Perhaps surprisingly, however, *such systematic assessments are missing from current literature*.

**Research Gap.** Scientific literature on phishing defense can be divided in two categories: *technical papers* (e.g., [10, 48, 49, 51, 52]), which propose (or attack) a given solution; and *user studies* (e.g., [8, 34, 80]), which seek to investigate the response of humans to phishing (useful for phishing training and education). However, to the best of our knowledge, none of these categories have questioned how humans respond to phishing webpages crafted to bypass ML-PWDs. Indeed, from an "adversarial ML" perspective, technical

papers typically stop after showing that a given ML-PWD has been evaded (e.g., [11, 56]); whereas user studies either entailed "phishing" webpages that have been crafted ad-hoc (e.g., [34, 57]) or, even when real phishing webpages were considered (e.g., [8, 12]), the role of ML was irrelevant. Hence, the question: *"Are adversarial webpages a problem in reality?"* is still open. As a matter of fact, recent findings [9] revealed that the ML-PWD of a security company had over 9k false negatives in one month—some of which entailed "perturbations" that most laymen would notice (see Fig. 5).

**Related Work.** We acknowledge, however, that the limitations of prior work are well-justified. Indeed, technical papers can be complex, and carrying out user studies *on top* of devising a scientifically sound and relevant contribution is challenging; whereas user studies require the availability of ML-powered PWD, which are becoming popular only in recent years. Nonetheless, we found *two works which partially overlap* with ours. **(1)** Abdelnabi et al. [6], after proposing an ML-PWD, discuss a user study (in the Appendix, with limited details) wherein participants were shown the webpages that bypassed the proposed ML-PWD and asked to rate "how trustworthy" such webpages were. The purpose of the user study, however, is to assess user agreements with their proposed similarity metric, and thus it does not involve the assessment of adversarial phishing pages or their comparison with benign/unperturbed phishing pages. **(2)** Lee et al. [48] attack an ML-PWD which exclusively focuses on the logo of well-known brands, and then carry out a user study asking participants how similar an adversarial logo was w.r.t. an original logo: the problem is that the logo is only a single element in a webpage (i.e., the webpage could be still detected by other automated mechanisms).

**Our Goal.** In this paper, we seek to overcome the shortcomings of prior work. Specifically, we investigate the response of human users to "adversarial" phishing webpages[1] that evaded ML-PWD (both real ones and custom-made); then, we compare such results with the ones from user assessments of "non-adversarial" phishing webpages. The rationale is that attackers are less interested in crafting adversarial webpages that, despite evading ML-PWD, can be easily spotted by end-users—i.e., their final target.

## 3 DATA COLLECTION & GENERATION

To answer our research questions, we design user studies wherein participants are asked to examine a mixed set of phishing and legitimate webpages. A crucial part of our research is that we want to investigate the response of users to adversarial webpages that *bypassed* ML-based detectors (both synthetic ones, as well as real products); indeed, this is necessary to determine whether adversarial webpages represent a problem "in reality". Therefore, before describing our user studies, we explain how we obtained a set of adversarial webpages that we can use for our user studies. Fig. 1 summarizes the workflow of our experimental methodology.

**Overview.** We seek to identify adversarial webpages that bypass either production-grade ML-PWD, or state-of-the-art research proposals. To meet this twofold requirement, we must first obtain a dataset including both benign and phishing webpages—which

---

[1]We **focus on phishing "on the Web"**. Other forms of phishing (such as via email [67] or phone calls [14]) and their detection (with or without ML) are orthogonal research areas to this paper (albeit some of our findings can be relevant also to these areas).

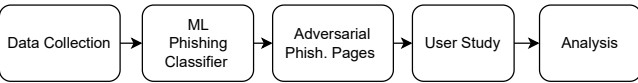

Fig. 1: Workflow of our study.

| Brands |
|---|
| Adobe, Amazon, Apple, AT&T, Bank of America, DHL, Dropbox, eBay, Facebook, Google, Microsoft, Outlook, Paypal, Wells Fargo, Yahoo |

Table 1: We selected 15 brands, popular in the U.S., for our user study.

will be used to develop a custom ML-PWD. Then, after ensuring that our ML-PWD obtains good detection performance (i.e., high true positive rate with low false positive rate) in "non-adversarial" scenarios, we will use the phishing webpages in our dataset as the basis to craft adversarial phishing webpages. Such adversarial examples will then be tested against our custom ML-PWD. If they can evade the detection, we will consider them for our user study.

**Dataset.** To develop a state-of-the-art ML-PWD, we rely on the phishing dataset by Chiew et al. [23]. This dataset (used also, e.g., in [65]) contains 30k webpages: 15k are benign (source: Alexa top) and 15k are phishing (source: Phishtank [3]). We consider this dataset because, for each sample, it provides the HTML content as well as supporting files (e.g., CSS) and all the image components. This allows us to craft *realizable* perturbations on these webpages, thereby yielding adversarial webpages with high realistic fidelity. Other existing datasets (e.g., [11, 52]) do not allow this, since they lack CSS and/or image files. Finally, although our chosen dataset was released in 2018, its webpages still resemble the ones of the "current" version (as of Sept. 2023) of the corresponding websites.

**Custom ML-PWD.** We first use the dataset [23] of benign and phishing webpages to train a ML-PWD. Then we add perturbations to a phishing webpage, aiming to trigger a false negative by the ML-PWD. In more detail, our ML-PWD relies on the random forest algorithm (thanks to its superior performance over other ML algorithms, as reported by many prior works [10, 75]). In particular, we rely on the code (and features) provided by [10] to develop our ML-PWD, for which we use 80% of the dataset for training and use the remaining 20% for testing. Our ML-PWD obtains performance comparable with the state-of-the-art, having a true positive rate of 0.98 and a false positive rate of 0.04 (results aligning with prior works [10, 65]). These results confirm that our ML-PWD (which we release [1]) *is a valid candidate for our research.*

**Custom Adversarial Phishing Webpages.** We use/adapt existing AML methods (borrowed from [10] and [82]) to generate 4 types of adversarial phishing webpages "in a lab" (*APW-Lab*). More specifically, we selected *four* types of perturbations that help a phishing page evade our custom ML-PWD, each yielding an adversarial phishing webpage having diverse visual cues:

(1) *APW-Lab_img*: we insert a small array of images to the bottom of the web page (footer), as shown in Fig. 2(a).
(2) *APW-Lab_typo*: we randomly insert typos to the text content of the web page as shown in Fig. 2(b).
(3) *APW-Lab_pswd*: we make the password visible for the password input box, as shown in Fig. 2(c).
(4) *APW-Lab_bg*: we randomly add a background image to the web page, as shown in Fig. 2(d).

The *APW-Lab* that bypass our ML-PWD will be used for the user study. We note that related work from Lee et al. [48] did not evaluate webpages but focused on logos only.

**Real Adversarial Phishing Webpages.** A prior work [9] identified 100 adversarial phishing websites "from the wild Web" that bypassed a production-grade ML-PWD (reliant on visual similarity) in July 2022. A close inspection shows that these adversarial pages adopt various evasion strategies such as using blurry logos and adding background patterns (example in Fig. 6 in the Appendix). We will use this set (denoted as *APW-Wild*) to examine the user perception on adversarial webpages crafted by real phishers (we note that neither Lee et al. [48] nor Abdelnabi et al. [6] considered real phishing webpages that bypassed a production-grade ML-PWD).

## 4 USER STUDY: SET-UP

We carry out two user studies. The first, serving as a baseline, examines how well users can distinguish legitimate webpages from "unperturbed" phishing webpages. The second examines how well users can distinguish "adversarial" phishing webpages (APW) from legitimate ones. Henceforth, we refer to the first user study as *baseline study*, and to the second as *adversarial study*.

### 4.1 Candidate Webpages

**Considered Brands.** To conduct a meaningful research, we only consider webpages representing a limited set of well-known brands.[2] Hence, we select the 15 well-known brands (typically targeted by phishing attacks [5]) shown in Table 1.

**Webpage Classes** For these selected brands, we construct a user study dataset spanning the following classes of webpages:
- *Legitimate.* For each brand in Table 1, we retrieve the (legitimate) webpage corresponding to the brand's homepage.
- *Unperturbed Phishing.* For each brand, we randomly sample two phishing webpages from our chosen dataset (cf. §3).
- *APW-Lab.* For each brand and perturbation type, we select one adversarial webpage that bypassed our ML-PWD.
- *APW-Wild.* From the 100 webpages collected by Apruzzese et al. [9], we find 28 of them matching 8 of our target brands (i.e., Apple, AT&T, DHL, Dropbox, Google, Microsoft, Outlook, and Paypal), hence we randomly draw from these 28. We show some examples in Appendix A.

Overall, our user studies entail 15 legitimate, 30 unperturbed phishing webpages, 60 *APW-Lab* webpages, and 28 *APW-Wild* webpages.

### 4.2 Questionnaire Design

Both of our user studies are designed as questionnaires following a similar structure, depicted in Table 2. In what follows, we describe this common user study process from a participant's perspective.

**General Procedure.** At a high-level, the questionnaires consist of three parts. **(1)** A participant starts by reading a consent form stating their rights and the study's objectives. Afterwards,

---

[2]Indeed, some users may not be familiar with some less-popular brands, and their responses would have limited value for the purpose of our RQ.

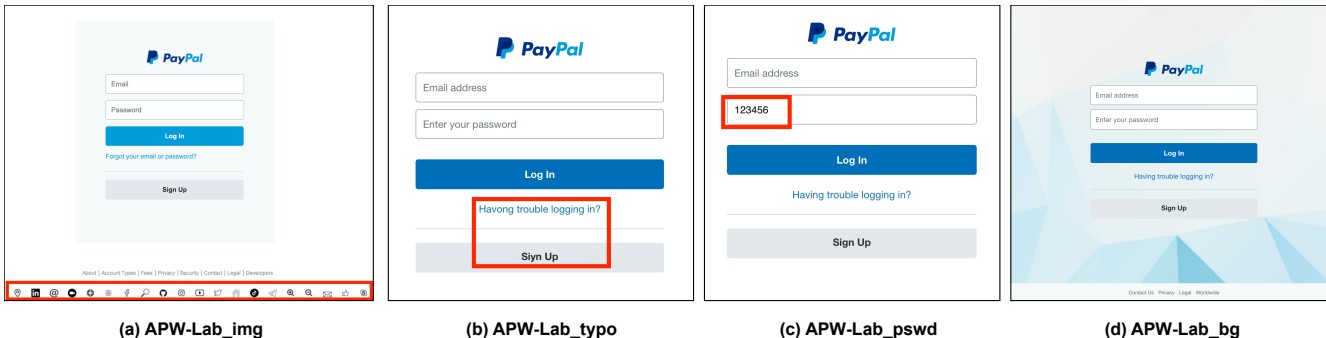

**Fig. 2: Example screenshot of lab-generated adversarial phishing pages targeting Paypal. We include two types of perturbations: (a) adding small images to the footer, (b) introducing typos, (c) making the password visible, and (d) adding a background image.**

| Study | Pages Seen by Each Participant | Participants |
|---|---|---|
| Baseline | 7 Legitimate + 8 Unperturbed Phishing | 235 |
| Adversarial | 7 Legitimate + 4 *APW-Lab* + 4 *APW-Wild* | 235 |

**Table 2: Summary of our user studies. We report the classes of webpages that *each participant views* and the number of participants.**

the participant reads a brief introduction about phishing attacks and phishing websites. We explicitly inform the participants that the study is about detecting phishing websites. This is considered a "highly-primed" setting, i.e., participants may be more prepared to detect phishing websites than they would in the real world. We use this setting to estimate the *upper-bound performance* of users. This effect has been shown in previous phishing studies (e.g., [38]) where highly prompted participants have a better phishing detection performance than unprompted participants. **(2)** Then, the participant will view a total of 15 webpages (as screenshots, taken in high resolution and tailored for desktop browsers), covering all 15 brands in Table 1. The participant is asked to assess the legitimacy of each shown webpage. For the baseline study, each participant will view 7 legitimate pages and 8 unperturbed phishing pages. For the adversarial study, each participant will view 7 legitimate pages, 4 *APW-Lab* (one for each perturbation type), and 4 *APW-Wild*. The webpages to present to each user are randomly chosen, but we ensure the benign-to-phishing ratio and also that any given user will not see two (or more) screenshots of the same brand—thereby ensuring consistency, since all users will see 15 screenshots of 15 different brands). Furthermore, the order of the pages is randomized for each participant to avoid order bias [31] (this was not done by Lee et al. [48] or Abdelnabi et al. [6]). **(3)** Finally, the participant will answer some exit questions to report demographic information such as age, gender, education, and knowledge of phishing and the considered brands. For attention check, at the end of the main experiment we show a screenshot of a popular social network (Twitter/Instagram) and ask whether it represents a bank website.

**Detailed Questions.** Under each screenshot, we include two questions: "*How do you rate the legitimacy of this webpage?*" [Q1], and "*What specific components/indicators on the webpage have influenced your choice?*" [Q2]. For Q1, the participant is asked to rate the legitimacy of the web page from 1 to 6: 1 (definitely phishing), 2 (very probably phishing), 3 (probably phishing, but not sure), 4 (probably legitimate, but not sure), 5 (very probably legitimate) and

6 (definitely legitimate). The six-point Likert scale does not include a "neutral" option to encourage participants to draw a conclusion. For Q2, the participant provides open-ended answers via a text box.

For the exit questions, we first inquire the participant's familiarity with the considered brands—"*Do you know these brands/companies/services?*" and "*Please rate how often you visit the websites of these brands*". The participant provides a binary answer for the first question and uses a 4-point Likert scale for the second question. Then, we ask the participant about their gender, age, highest education level, and whether they have a technical background in cybersecurity. More details about these questions are in Appendix B.

### 4.3 Recruitment, Ethics, and Demographics

Our study was reviewed and approved by our IRB; we also follow the Menlo report [15] and do not deploy any phishing webpage on the Web (we only show screenshots). We recruited participants from *Prolific* between July and August of 2023. We choose Prolific over other platforms such as MTurk for the high-quality work from Prolific [60]. Participation in our study is anonymous and voluntary, and participants have unlimited time to read the consent form. Participants can withdraw their consent at any time without any risk. We did not collect any personally identifiable information [42], nor sensitive data [4]. Considering that our target brands are mostly U.S.-based websites, we focus on participants from the U.S. from Prolific. After filtering out low-quality answers (based on attention check), our sample[3] encompasses *n*=470 participants (235 for each study). The age distribution ranges from 18 to 70+, with 240 males and 220 females (6 non-binary and 4 prefer not to say). Each participant can only join once and receive $2.2 compensation. On average, each participant spent 18.1 minutes on each questionnaire.

## 5 DETECTION RESULTS (QUANTITATIVE)

We first focus on answering RQ1–RQ3. To this purpose, we perform a *quantitative analysis* of the responses we collected for our two user studies. We begin by reporting the results at a high-level (§5.1), and then perform formal regression analyses (§5.2 and §5.3) to assess the statistical significance of our observations.

---

[3]Our user studies have a **population that is larger** than most previous user studies on (non-adversarial) phishing webpages [16]. Specifically, most works ([7, 8, 12, 13, 26, 37, 44, 45, 66, 67, 81]) have less than 100 participants, while five ([34, 41, 57, 76, 80]) have [100–400] participants. Only the work by Purkait et al. [63] has more participants (621) than ours, but it was carried out in 2014.

## 5.1 Overview (how good are our respondents?)

We report the overall performance of both user studies in Fig. 3, showing how well our participants correctly recognized each web-page.[4] By comparing the results of the two user studies (useful for RQ1), we observe that our participants exhibit a similar performance in identifying *legitimate* webpages (86% for the baseline study, and 88% for the adversarial study). In contrast, and perhaps worryingly, we found that their ability to recognize *phishing* webpages is much worse; intriguingly, however, it appears that our respondents can more easily discern adversarial phishing webpages (62%) than "unperturbed" ones (51%).

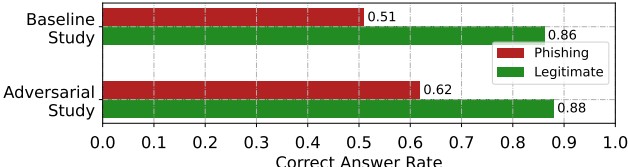

**Fig. 3: Overview of baseline and adversarial study (7,050 responses)**

In Fig. 4, we focus on the detection rates for *phishing* webpages. Specifically, we break down the results for the *adversarial* phishing webpages (*APW-Lab* and *APW-Wild*) and compare them with the "unperturbed" ones of the baseline study (useful for RQ2). This more detailed comparison reveals that our respondents are not easily tricked adversarial perturbations entailing 'typos' (i.e., the detection rate for *APW-Lab_typo* is 85%). However, they appear to be unable to spot other types of perturbations (i.e., the detection rate for the other three types of *APW-Lab* ranges between [50–56%]). Finally, the detection rate of *APW-Wild* aligns with the general trend (63%), suggesting that adversarial webpages "from the wild Web" are less effective at fooling real users.

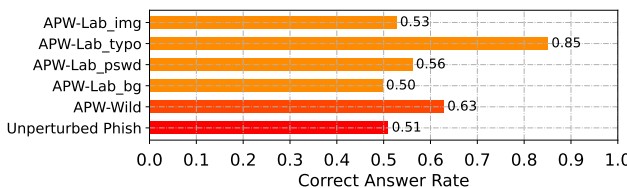

**Fig. 4: Detection rate for different types of phishing webpages.**

> **Observations:** (1) Our respondents can be deceived by phishing webpages. (2) Some adversarial perturbations are easy to spot by humans. (3) Adversarial webpages from the real world are less effective than "unperturbed" phishing webpages.

## 5.2 Statistical Analysis: Websites (RQ1 and RQ2)

To answer RQ1 and RQ2, we perform a rigorous analysis to ascertain the statistical significance of our previous findings.

**Method.** We choose a *mixed-effects logistic regression model* (used in many similar studies [16, 81]) to model the process of a user classifying a given webpage. The dependent variable ($y$) is *the correctness of the user's classification result for this webpage*. The

---

| Variable | Estimate ($\beta$) | Std. Err. | p-value |
|---|---|---|---|
| *Intercept* | 0.161 | 0.146 | 0.271 |
| Website type: Reference = Unperturbed Phishing | | | |
|     Legitimate | 1.912 | 0.073 | **<0.001***** |
|     *APW-Lab_img* | 0.049 | 0.144 | 0.734 |
|     *APW-Lab_typo* | 1.723 | 0.193 | **<0.001***** |
|     *APW-Lab_pswd* | 0.185 | 0.145 | 0.202 |
|     *APW-Lab_bg* | -0.075 | 0.144 | 0.605 |
|     *APW-Wild* | 0.484 | 0.089 | **<0.001***** |
| Knowledge of Website: Reference = NO | | | |
|     YES | -0.034 | 0.145 | 0.812 |
| Frequency of Visiting: Reference = Rarely or Never | | | |
|     Sometimes or Frequently | -0.169 | 0.059 | **0.004**** |

**Table 3: Webpage Classification Analysis** – Logistic mixed-effects regression model: we predict whether a website is classified correctly by a user, based on the type of website, the user's knowledge of this website/brand, and the user's frequency of visiting the website. Statistical significance is denoted by *** ($p < 0.001$), ** ($p < 0.01$), and * ($p < 0.05$) [25].

answer is coded as "1" if the classification is correct, and "0" otherwise. We model webpage types and user familiarity with the brand as *fixed effects* (independent variables). We treat each participant as a *random effect* because the same user has viewed 15 webpages (i.e., repeated measures). In this model, we have 3 independent variables ($x$) related to the webpage: (1) webpage type, (2) the user's prior knowledge of this webpage's brand, and (3) the user's frequency of visiting webpages of this brand. We include (2) and (3) for a simple intuition: if a user is familiar with a brand and visits its webpages regularly, they would be well-acquainted with its typical appearance, and thus are more likely to have a better detection accuracy. For webpage type, we have 7 types, and we treat "unperturbed" phishing webpages as the reference to compare with other 6 types. For knowledge of the website, we code the answer into a binary format and use "No" as the reference. For the website visit frequency, we also code the answer into a binary format and use "Rarely or Never" as the reference.

**Results.** The model is summarized in Table 3. We report standard metrics including *Estimate*, *Standard Error.* and *p-value* for the hypothesis tests. *Estimate* ($\beta$) describes the estimated effect of each predictor variable on the dependent variable while holding all other predictor variables constant. The sign of Estimate indicates the direction in which the dependent changes with the independent variables. A positive sign means that as the independent variable increases, the dependent variable also increases; otherwise, the dependent variable decreases. *Std. Err.* represents the average distance that the observed values fall from the regression line. The *p-value* in the regression model describes whether the relationships observed in the samples by chance; usually, the influence was considered statically significant when $p<0.05$.

**Analysis.** The results in Table 3 confirm our earlier observations from descriptive statistics. First, w.r.t. "unperturbed" phishing webpages, we find that legitimate webpages are statistically significantly easier to detect ($\beta$=1.912, $p<0.001$). Second, among the adversarial webpages, we find two types that are statistically easier to detect by users: *APW-Lab_typo* ($\beta$=1.723, $p<0.001$), indicating that even though the typo is subtle, it has raised suspicion of users; and

*APW-Wild* ($\beta$=0.484, $p$<0.001), revealing that while some adversarial webpages from the wild Web can bypass production-grade ML-PWD, they indeed make users more suspicious (w.r.t. "unperturbed" phishing pages). Finally, we did not find statistically significant differences between "unperturbed" phishing webpages and other types of APW. These include adversarial phishing webpages with image footers (*APW-Lab_img*), or visible passwords (*APW-Lab_pswd*), or with changed background images (*APW-Lab_bg*): all these APW can bypass state-of-the-art ML-based detector and yet do not raise more suspicion from users' perspectives.

Table 3 also shows an intriguing phenomenon regarding how users' familiarity with the brand correlates with their detection performance. First, we did not find statistically significant evidence that users' prior *knowledge of a brand* influences their detection. However, users' *frequency of visiting the brand's webpages* has a statistically significant *negative* correlation with their detection correctness ($\beta$=−0.169, $p$=0.004). In other words, users are more likely to make incorrect guesses about webpages of brand that they visit "sometimes or frequently", compared with another that they "rarely or never" visit. This may suggest that familiarity with the brand could lead to overconfidence, i.e., where one's judgmental confidence exceeds one's actual performance in decision-making [62, 78].

---

**TAKEAWAYS** (RQ1-2): We make four statistically significant findings. From a user perspective, compared to "unperturbed" phishing webpages: **(1)** adversarial phishing webpages with typo-based perturbations are easier to detect; **(2)** adversarial phishing webpages found in the wild Web are more recognizable; **(3)** adversarial perturbations such as inserting images to the footer, making the password visible, or adding a background image, do not make phishing webpages more suspicious. Finally, **(4)** users are more likely to misdetect webpages that they visit more frequently.

---

## 5.3 Statistical Analysis: Users Attributes (RQ3)

We now turn our attention to RQ3, and rigorously examine how users' attributes influence their phishing detection performance.

**Method.**     We construct a user model using a *linear regression model* (used in many related studies [16, 63]). The dependent variable is a user's correct answer rate (i.e., accuracy) among the 15 pages they viewed. The independent variables include various user attributes such as demographic factors, technical backgrounds, knowledge of phishing, and time spent on the survey. We code the independent variables in a binary format, except for the time spent on the questionnaire (which is numerical).

**Results and Analysis.**     We display the results in Table 4, showing the absence of statistically significant evidence that users' demographic factors affect their phishing detection performance. Instead, a user's prior knowledge of phishing has a statistically significant influence. More specifically, users with prior knowledge of phishing are more likely to achieve a higher detection accuracy ($\beta$=0.036, $p$=0.008). Even though the estimate $\beta$ is small, the difference is statistically significant. Our result (in the context of *adversarial* webpages) is slightly different from prior user studies on phishing [33, 38, 44, 63, 71] wherein researchers found that demographic

| Variable | Estimate ($\beta$) | Std. Err. | p-value |
|---|---|---|---|
| *Intercept* | 0.693 | 0.018 | <0.001*** |
| Gender: Reference = Female | | | |
| Male | -0.001 | 0.013 | 0.964 |
| Age: Reference = Younger (<= 39) | | | |
| Older (>39) | -0.004 | 0.012 | 0.751 |
| Education: Reference = Lower (< Bachelor) | | | |
| Higher (>= Bachelor) | -0.004 | 0.013 | 0.783 |
| Phish knowledge: Reference = NO | | | |
| YES | 0.036 | 0.013 | **0.008**** |
| Computer knowledge: Reference = NO | | | |
| YES | 0.029 | 0.019 | 0.122 |
| Security knowledge: Reference = NO | | | |
| YES | -0.003 | 0.029 | 0.931 |
| Time Spent on Survey | -0.001 | 0.001 | 0.293 |

**Table 4: User Attribute Analysis** – Linear regression model: we predict a user's detection accuracy based on the user's attributes such as demographic factors, technical background, and knowledge of phishing. Statistical significance is denoted by *** ($p < 0.001$), ** ($p < 0.01$), and * ($p < 0.05$) [25].

factors such as gender or age have influenced users' detection performance. Finally, the time a user spent on the survey does not seem to have a significant influence on the user's detection accuracy.

---

**TAKEAWAYS** (RQ3): We did not find statistically significant evidence that demographic factors affect users' detection accuracy. A user's prior knowledge of phishing is a significant predictor.

---

## 6 USERS' REASONING (QUALITATIVE)

We now address RQ4. Recall (see §4.2) that, for every webpage shown in the questionnaire, we also asked (with [Q2]) participants (P) to point out the cues that influenced their rating (of [Q1]). Here, we qualitatively analyze the open-form answers through a *thematic analysis* [73] (which has been used also in [9]).

**Codebook.**     Given that we focus on adversarial phishing webpages, our qualitative coding is based on the data from the adversarial study. In total, we have 3,525 responses from 235 participants from the adversarial study. Two authors (i.e., coders) have worked together to code the answers. A primary coder first codes 27% of the responses, which serves as the foundation for creating a comprehensive codebook. Subsequently, both the primary and secondary coders independently code 10% of the responses that have not yet been coded. We use Cohen's Kappa ($\kappa$) statistic to assess the agreement between coders. In cases where $\kappa$<0.7, both coders meet up to discuss and resolve discrepancies and refine the codebook, potentially also re-examining and re-coding responses that exhibit inconsistencies. This iterative process continues until a satisfactory agreement is reached, i.e., $\kappa$>0.7 [55]. In our finalized codebook, we have $\kappa$=0.718, indicating *good inter-coder reliability* [32]. With this codebook (which we release [1]), we thematically coded 1,307 valid responses (37%) that mentioned any webpage elements [9] (e.g., logo, background) or their feeling of the webpage. Specifically, 737 responses are from webpages rated as "phishing" and 541 responses are from webpages rated as "legitimate".

## 6.1 Why is the webpage legitimate/phishing?

We first investigate what led our participants to derive that a given webpage is legitimate or phishing. For the sake of this analysis, we ignore the ground truth of each webpage, since we are interested in the users reasoning of what *they think* is phishing (or not).

**"I think this is Phishing because..."** Among the 737 responses on webpages rated as phishing, the most prevalent factor is "text content" (282, 38%). Other top-3 factors are "layout" (170, 23%) and "functionality" (168, 23%) of the webpage. Fewer responses (66, 9%) mentioned image content. (We omit factors whose prevalence is below 9%.) We run pairwise Chi-squared tests to compare the number of responses mentioning *text content* (the most prevalent) and those mentioning each of the other factors. We confirm that the differences are statistically significant (all comparisons have *p*<0.001).

Among the 282 **text-related** responses, 119 of them (42%) mentioned the presence of typos. For example, P404 stated "*The spelling of the word Outlook is not right*". This is consistent with prior studies [29, 53] reporting that typos hurt the perceived credibility of a webpage. Other text-related responses encompassed factors such as "grammar" (67, 24.5%) and "style" (44, 15.6%). E.g., P1013 mentioned "*The font does not look like the regular Google font that I usually see*".

Regarding other prevalent factors, **layout** (23%) refers to the organization of different components of the webpage, which is a known factor that influences the perceived credibility of websites [18]. E.g., P496 stated "*This does not look like the regular Google login page at all; it looks really off so it seems super sketchy.*" The **functionality** (23%) denotes the specific tasks that the website can help users to accomplish. E.g., P520 mentioned "*This does not appear to be a correct website for DHL since they would not ask you to log in typically to track*". Nonetheless, participants expected that phishing websites would have a way to collect user data. As such, such information-gathering functionality can raise suspicion. E.g., P825, in response to the page shown in Fig. 7 (Appendix A), stated "*it asked for the credit card number and therefore looks like it phishing*".

In comparison, fewer responses mentioned **image** content (66, 9%). E.g., P860 mentioned "*The image seems off from what I am usually used to*". Among these, 25 responses mentioned the background, e.g., P1202 stated "*The background isn't moving like on the real site*".

**"I think this is Legitimate because..."** Among the 541 responses for webpages rated as legitimate, 249 (46%) did not mention any specific factor but describe how the participant "feels" about the webpage. E.g., P154 stated: "*(It) looks like PayPal login page*". Only few responses mentioned specific factors. E.g., 26 (5%) mentioned "*no misspellings or poor grammar*", suggesting that correct writing is regarded as an indicator of legitimacy (albeit this could be influenced by previously viewed webpages having typos). Finally, we report that some users may rely on misinformed strategies. E.g., P54 stated: "*Google is a very reputable and credible search engine*", suggesting that a brand's reputation is an indicator of trustworthiness (which is exactly what phishers use to trick their victims).

> **Takeaways** (RQ4): After determining the legitimacy of a webpage, users motivate their decision by describing their "feelings" if they believe the webpage to be legitimate. In contrast, when they think the webpage is phishing, they mention more specific indicators—most of which entail textual content errors.

## 6.2 What do users write on adversarial samples?

In an attempt to exhaustively answer RQ4, we further enrich our analysis by performing a break down of the participants' reasoning on the *specific type* of APW (cf. §3) included in our adversarial study. For this investigation (and contrarily to what we did in §6.1), we must account for the ground truth of each webpage.

**APW-Lab.** We recall (cf. Fig. 4) that our participants performed very well on *APW-Lab_typo*, for which we coded 93 responses. Among these, a large majority (69, 74%) mentioned "typo" (after making a correct detection). Intriguingly, 15% (14) provided reasons that have nothing to do with *APW-Lab_typo* (despite still rating them as phishing). E.g., P668 stated: "*figures do not look normal*". The remaining 11% incorrectly labeled the webpage as legitimate (e.g., "*Everrything looks normal*" [P621]).

Concerning *APW-Lab_img*, we have coded 61 responses. Notably, only 13% (8) pointed out the 'correct' adversarial perturbation (i.e., images on footer). E.g., P544 stated: "*low quality and strange icons at the bottom, which a legit site would not have*". In contrast, 48% (29) mentioned other reasons. E.g., P210 stated: "*Adobe doesn't require logging in to view something in it to my knowledge*". The remaining 39% incorrectly labeled the webpage as legitimate (e.g., "*norton certificate makes me think it's more legit than not.*" [242]).

For *APW-Lab_pswd*, we coded 137 responses. The majority (70, 51%), despite stemming from a correct detection, have nothing to do with our perturbation: only 8% (11) pointed out the visible password as a potential phishing indicator (e.g., "*password field is plain text*" [P1306]; or "*the password is not hidden*" [P937]). The rest 41% incorrectly labeled the webpage as legitimate (e.g., "*As a Wells Fargo customer who was literally just checking their account before starting this study I can assure you you is legitimately legit*" [P86]).

We coded 89 responses for *APW-Lab_bg*. Surprisingly, only 4% (3) of responses mention our inserted perturbation. In contrast, 48% (43) justify their (correct) phishing detection by mentioning unrelated factors. E.g., P971 stated: "*too many big competing brands at the top*". The rest 49% incorrectly labeled the page as legitimate (e.g., P321 stated: "*good grammar, good syntax, appropriate colors, logo*").

For each type of APW above, we again run a Chi-squared test to compare the number of correct phishing detections that mention the inserted perturbation w.r.t. other factors (we do not include misclassifications). The results show that the number of mentions of inserted perturbations is statistically significantly lower than other factors, with *p*<0.001 for all four perturbation types.

> **Takeaway** (RQ4): Even though participants can recognize an APW as "phishing", they rarely pinpoint the perturbation that makes the webpage "adversarial" (as long as it is not text-based).

**APW-Wild.** We coded 594 for adversarial webpages "from the wild Web". We recall (§5) that our participants are better at detecting *APW-Wild* (w.r.t. unperturbed phishing webpages), so we attempt to find an explanation for this. Driven by our previous findings (§6.1), we scrutinized whether the reason lies in text-related factors. Among the justifications for correct detections, we found that 22% (131) mention text-related factors (e.g., P1246 wrote "*'Forgotten password' doesn't seem right*"). More specifically, the responses mention typo, grammar, and text-style issues 8%, 6%, and 6%, respectively. Some (18%, 107) mentioned layout (e.g., P362 wrote "*bad*

*css*"), whereas others (16%, 94) mentioned functionality (e.g., P795 wrote: "*(It) should be one form of 2FA*"). Few 9% (56) mention the logo (e.g., P1007 wrote "*The Google logo is wrong.*"); and even less (7%, 40) mentioned other visual elements such as background color (e.g., P108 wrote: "*Google login prompt is not with a gray background*"). Finally, 205 (35%) incorrectly labeled ther webpage as legitimate (e.g., "*Nothing misleading*" [P119]). We run a Chi-squared test, and confirm the number of mentions of text indicators is higher than functionality, logo, and other visual elements, with statistical significance ($p < 0.01$ for all pairs). However, the difference between text indicators and layout is not statistically significant ($p = 0.082$).[5]

## 7 DISCUSSION

**Comparing with Prior Phishing Research.** Our work examines how users perceive *adversarial phishing webpages*, which has never been studied in prior works. This provides an interesting data point to contrast with prior studies on generic phishing websites and emails [16]. We discuss four points. **(1)** Prior studies show that men perform better on phishing detection tasks (website [39, 44], email [78, 79]) and a few studies show that women perform better (website and email [59]). Our analysis does not find statistically significant differences among genders (§5.3). **(2)** Prior studies show that elders are more susceptible to phishing websites [44, 71]. We again do not find statistically significant differences with respect to age groups (§5.3). **(3)** Our study echoes prior results that phishing knowledge correlates positively with users' phishing detection performance [28]. However, surprisingly, we find that the frequency of a user visiting a target brand's website *negatively* correlates with the user's ability to detect phishing webpages targeting this brand (§5.3). An explanation is that "familiarity with a brand" leads to overconfidence [62, 78]. This may align with the prior observation that people feel more comfortable with (i.e. trusting) websites that they are familiar with [74]. **(4)** Prior studies have independently shown that typos [33, 53], webpage layout [18], and webpage visual appearance [8] would influence the perceived credibility of websites (and unperturbed phishing webpages). Under the context of *adversarial* phishing, our study shows that participants are significantly more sensitive (§6.1) to adversarial perturbations related to typos and text in general (w.r.t. other visual perturbations).

**Implications for 'technical' Web Security.** For research focused on adversarial phishing attacks (e.g., [10, 24, 48, 49, 68]), we argue that bypassing a given ML-PWD is necessary but not sufficient for a phishing webpage to be successful. The adversarial phishing webpages should be also assessed with users. More importantly, it is important to compare adversarial phishing webpages with unperturbed phishing webpages to ensure the adversarial perturbations do not make the webpages significantly less effective on users (in favor of bypassing ML-PWD). E.g., in our study, we find that certain adversarial perturbations (e.g., typos) are more easily noticed by users despite their high evasion success rate against ML-PWD. This defect would be otherwise unknown without a user study. Another implication is that visual-based adversarial perturbations seem to be effective against both ML-PWD and users, which

should be considered in future work when robustifying ML-based phishing detectors. Finally, we stress that some of our visual perturbations were "large" (e.g., *APW-Lab_bg* entailed replacing the entire background—see Fig. 2), but they still allowed the webpage to bypass the ML-PWD (both ours and the production-grade one—see Fig. 5) *and deceive the users*. This is in stark contrast with most AML research in computer vision, wherein the goal is to apply "imperceptible" perturbations (e.g., [20, 70]). Hence, we endorse future research to explore perturbations having higher magnitude.

**Implications to User Education.** Researchers have studied ways to improve users' ability to recognize phishing websites through training and education [46, 57, 80]. Our results show that users overlook 'visual' adversarial perturbations (w.r.t. text-based ones). One possible future direction is to increase user awareness of such adversarial phishing webpages. However, we believe there is an inherent risk to train users to search for such visual artifacts. Indeed, adversarial phishing webpages have certain visual artifacts that deviate them from authentic phishing webpages—helping users recognize such artifacts may help users with phishing detection. However, *the lack of such artifacts* does not mean the website is trustworthy. In our study, we have observed signs of over-trusting known/familiar websites. For example, a user's frequency of visiting a brand's website negatively predicts the user's phishing detection accuracy on this brand.

**Limitations.** First, our study is limited to participants from the U.S. given we are primarily assessing phishing sites targeting the US-based brands. Future work may consider recruiting participants from different countries and expanding the set of target brands. Second, our evaluation is intentionally set to be highly primed to examine the upper-bound performance of users. This can be different from real-world scenarios wherein users are often "unprepared" when encountering phishing websites. Third, to protect users, we only present phishing screenshots (to prevent users from accidentally clicking on malicious links or leaking their information). However, this also prevents interacting with the website which can be a part of the human's detection process. Furthermore, our screenshots are for desktop browsers, and hence we do not claim that our results generalize to other platforms (e.g., smartphones). Finally, to focus on adversarial phishing webpages, we excluded URLs from our evaluation. Even though prior studies [26, 50, 80] showed that most users cannot effectively utilize URLs as identity indicators of a website, the presence of URLs may help users judge the overall legitimacy of a webpage together with other indicators.

## 8 CONCLUSION

We present two user studies ($n = 470$) to assess how human users perceive adversarial phishing webpages that bypass ML-based phishing website detectors. We confirm the threat of adversarial phishing webpages to end-users and compare the effectiveness of different types of adversarial perturbations. We argue that *assessing the users' response to adversarial webpages* should be a mandatory step to evaluate evasion attacks in the context of phishing webpage detection. Our work can serve as a benchmark for future research, and we release our questionnaires, codebook, classifiers, and datasets [1].

---

[5]We refrain from making claims pertaining the "correct identification" of the perturbation (as we did for *APW-Lab*): this is because we cannot be sure of which perturbation was applied by the (real) attackers who crafted the webpages in *APW-Wild* [9].

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

# A  SUPPLEMENTARY FIGURES AND TABLES

## A.1  Number of Experimental Webpages

Our user study involves 15 well-known U.S. website brands. As illustrated in Table 5, for each brand, we have 2 high-quality unperturbed phishing pages, 1 legitimate webpage, 4 types of *APW-Lab* pages, and a variable number of *APW-Wild* pages ranging from 0 to 7.

| Brand | APW-Lab | APW-Wild | Unperturbed Phish. | Legitimate |
|---|---|---|---|---|
| Adobe | 4 | 0 | 2 | 1 |
| Amazon | 4 | 0 | 2 | 1 |
| Apple | 4 | 2 | 2 | 1 |
| AT&T | 4 | 7 | 2 | 1 |
| Bank of America | 4 | 0 | 2 | 1 |
| DHL | 4 | 2 | 2 | 1 |
| Dropbox | 4 | 2 | 2 | 1 |
| eBay | 4 | 0 | 2 | 1 |
| Facebook | 4 | 0 | 2 | 1 |
| Google | 4 | 7 | 2 | 1 |
| Microsoft | 4 | 4 | 2 | 1 |
| Outlook | 4 | 3 | 2 | 1 |
| Paypal | 4 | 1 | 2 | 1 |
| Wells Fargo | 4 | 0 | 2 | 1 |
| Yahoo | 4 | 0 | 2 | 1 |

**Table 5: Number of Experimental Webpages**

## A.2  Additional Example Screenshots

Fig. 5 presents four adversarial phishing webpages in [9] that evaded production-grade ML-PWD. Fig. 6 shows two *APW-Wild* pages used in our study with a weird background pattern and a blurry logo. Fig. 7 is an adversarial phishing webpage (*APW-Wild*) that asks for credit card information.

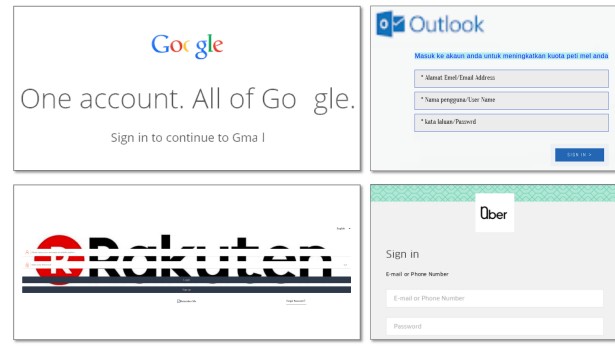

**Fig. 5: Four phishing webpages deployed "in the wild" (taken from [9]) which bypassed production-grade ML-PWD.**

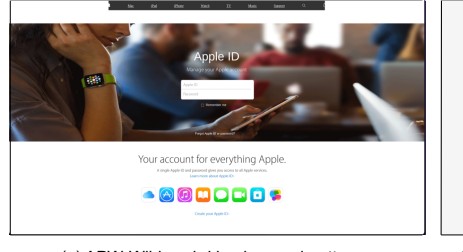

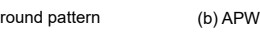

(a) APW-Wild: weird background pattern       (b) APW-Wild: blurry logo

**Fig. 6: Additional screenshot of APW-Wild pages used in our user study, to illustrate different adversarial perturbations.**

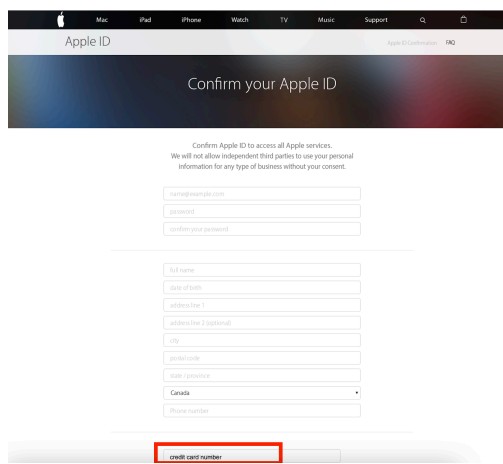

**Fig. 7: An adversarial phishing page asking for credit card information.**

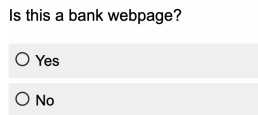

**Fig. 8: Attention check question.**

## B  STUDY QUESTIONS

In this section, we show a complete list of our questions, which includes the main task questions and other questions (website knowledge and demographic questions).

Each participant is instructed to review 15 webpage screenshots. Under each webpage, the participant answers 2 questions (15×2=30 questions in total), as shown in Fig. 10. Then, we randomly display the screenshot of Instagram or Twitter and show an attention check question (Fig. 8). After that, each participant needs to answer 2 questions about website knowledge (familiarity and frequency), as shown in Fig. 11 and 6 demographic questions, as shown in Fig. 9.

## C  ADDITIONAL BACKGROUND: PHISHING WEBSITE DETECTION AND ML SECURITY

Phishing websites are a never-ending problem that continue to pollute the Web, and rule-based countermeasures, such as blocklists, cannot cope with such a threat [58]. To provide some form of protection against "novel" phishing websites, modern anti-phishing schemes leverage data-driven techniques [75], such as machine learning (ML). Indeed, thanks to the capability of ML models to "automatically learn from data", it is possible to develop phishing website detectors (PWD) that can identify (and, consequently, block) malicious webpages *before* they are displayed to the end-user—**the actual target of a phishing attack**.

**ML-PWD.**  A large body of scientific literature proposed ML-driven PWD (ML-PWD), which can analyze various data-types to discriminate benign from phishing webpages. For instance, some solutions analyze the underlying HTML of a given webpage [40], or the characters that compose its URL [77], or a combination of the two [10]. Finally, recent approaches rely on deep learning (DL)

Do you have a technical background in cyber security?

○ Yes

○ No

○ Prefer not to say

Do you have knowledge about Phishing websites?

○ Yes

○ No

○ Prefer not to say

Do you have a technical background in computer science or computer engineering?

○ Yes

○ No

○ Prefer not to say

What is the highest level of education you have completed?

○ Some high school or less

○ High school diploma or GED

○ Some college, but no degree

○ Associates or technical degree

○ Bachelor's degree

○ Graduate or professional degree (MA, MS, MBA, PhD, JD, MD, DDS etc.)

○ Prefer not to say

How old are you?

○ 18-29

○ 30-39

○ 40-49

○ 50-59

○ 60-69

○ 70 or above

○ Prefer not to say

What is your gender?

○ Male

○ Female

○ Non-binary / third gender

○ Prefer not to say

**Fig. 9: Other questions: demographics.**

to compute the visual similarity between two webpages [6], or some of its elements (such as the logo [51]). Due to the promising results of these defenses, *production-grade PWD now integrate some*

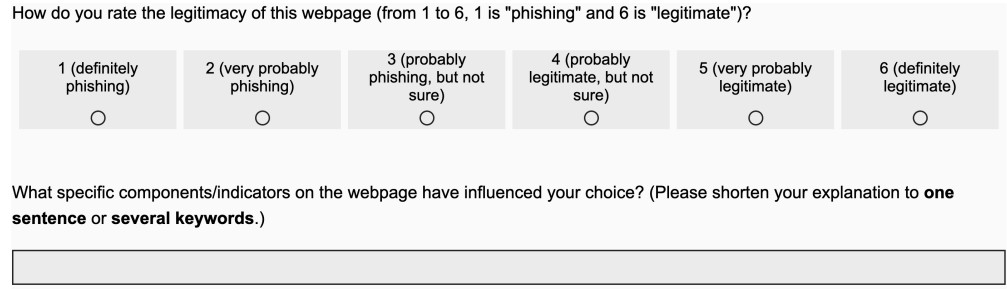

Fig. 10: Main task questions.

Fig. 11: Other questions: website knowledge.

*form of ML* to prevent their users from falling victim to a phishing hook [9, 27, 72].

**Security of ML.** The increasing (and not yet fully understood) successes of ML led to abundant papers to scrutinize its security [20] in adversarial environments. It is now well-known that ML-powered detectors are prone to *evasion attacks*, wherein (tiny) "adversarial perturbations" are added to a given input sample, so as to induce the detector to misclassify it—thereby triggering a **false negative**. Such a vulnerability has been investigated by thousands of research efforts [9], all of which showed that – no matter what – ML models can be easily bypassed (even "adversarially robust" ones [22]). Unfortunately, this problem also affects ML-driven PWD [10, 24, 48, 49]. For instance, some works (e.g. [68]) evidenced that the detection rate of some ML-PWD dropped from 95% to 0 by manipulating just a few features. Moreover, even production-grade ML-PWD exhibit the same weakness: both Google's [49] and BitDefender's [69] anti-phishing schemes have been defeated.

**Practitioners viewpoint.** Interestingly, however, there is abundant evidence showing that **ML developers do not have the ML-specific weaknesses among their priorities** [9]. Kumar et al. [43] did the first investigation on AML from the perspective of industry practitioners, which indicated only 5 out of 28 organizations had a working knowledge of AML. In the following year, [21]

investigated the current state of ML practitioners concerning ML security and privacy, and participants said "I Never Thought About Securing My Machine Learning Models". Even in the latest survey [35], only 28.7% of ML practitioners reported AML knowledge. Simply put, there is a clear gap between AML research and practice, which is not acceptable given the widespread deployment of ML into operational systems. Our paper seeks to rectify this mismatch—which, in the PWD context, presents intriguing properties that are currently overlooked.

