# OpenReview forum: "“Are Adversarial Phishing Webpages a Threat in Reality?” Understanding the Users’ Perception of Adversarial Webpages"
_ACM.org/TheWebConf/2024/Conference — TheWebConf24 Oral_

### Official Review · Reviewer_8d9d · 2023-11-20

**Novelty:** 5
**Technical Quality:** 4

**Review:**

The authors conducted two user studies involving 470 participants to assess human perception of adversarial webpages. The two user studies provided interesting findings about the relationship between user awareness and adversarial phishing webpages.

# Strengths

+ The paper stands out for its novel focus on user awareness of adversarial webpages, a subject rarely explored in existing research.

+ The research impresses with its large-scale approach, recruiting 470 participants for two separate studies.

+ One notable revelation is that significant visual perturbations can still evade ML-based phishing detection (ML-PWD), challenging prevalent assumptions in adversarial machine learning (AML) research.

# Weaknesses

Real-World Applicability: The study's focus on adversarial phishing webpages, which often exhibit notable content deviations (like added images or typos), raises questions about their practical relevance. Typically, phishing attempts exactly replicate legitimate webpages to deceive users. Traditional anti-phishing defenses, including URL scanning, SSL/TLS certificate validation, and DNS filtering, are primarily tailored to detect conventional phishing attempts. Given this, the paper's focus on adversarial phishing webpages, which deviate significantly from these standard methods, invites skepticism about its practical relevance. The study’s exploration into APWs, while academically intriguing, may not align with the typical defense strategies against more common phishing approaches.

The paper could benefit from a more thorough justification of the practical application of adversarial phishing webpages. Given that these APWs differ significantly from the typical phishing pages encountered in real-world scenarios, it's crucial for the study to delve deeper into how its findings can be applied effectively in the ongoing battle against phishing. This would bridge the gap between academic exploration and practical cybersecurity measures.

Participant Diversity Unclear: The absence of detailed demographic information on participants makes it difficult to assess the diversity and representativeness of the study sample.

Potential Bias in Experiment Design: The participants' awareness that they were part of a phishing detection study could have influenced their responses. A two-stage experimental design, initially concealing the study's focus, might yield more authentic results. In the initial stage, participants could engage with the experiment without any prior knowledge about its focus on phishing. This would likely result in more natural responses. In the subsequent stage, informing participants about the phishing element could provide contrasting insights into their detection capabilities.

Selection of Adversarial Webpage Types: The paper doesn't clearly justify the selection of specific adversarial webpage types (APWs) used in the study. It leaves readers questioning the representativeness and practical relevance of these chosen types. An explanation of why these particular APWs were chosen and how they relate to real-world phishing scenarios would add depth to the research and strengthen the study's conclusions.

Choice of Machine Learning Algorithm: The paper doesn't explicitly justify the use of Random Forest in the ML model, despite its mention in references. A more direct explanation would strengthen the methodology section.

**Questions:**

How do you envision the practical application of your findings on user awareness of adversarial webpages in real-world anti-phishing strategies, given the significant deviation of adversarial phishing webpages from the appearance of typical phishing pages?

Can you provide more information about the demographic diversity of the participant pool in your studies to ensure the representativeness of the findings?

Could the study's design, where participants were aware of its focus on phishing detection, have introduced bias?

On what basis were the specific types of adversarial webpages selected for the study, and how do they correlate with the phishing threats commonly encountered by users?

**Reviewer Confidence:**

4: The reviewer is certain that the evaluation is correct and very familiar with the relevant literature

**Scope:**

3: The work is somewhat relevant to the Web and to the track, and is of narrow interest to a sub-community

---

### Official Review · Reviewer_EQwi · 2023-11-21

**Novelty:** 5
**Technical Quality:** 5

**Review:**

This paper addresses the practical impact of adversarial phishing websites on users' perception, emphasizing the deceptive aspects faced by users rather than focusing solely on deceiving machine learning classifiers, as seen in prior literature. Through user studies, the authors contribute valuable insights, highlighting the potential threat posed by adversarial web pages to real users.

Strengths:
-	The study's motivation is robust.
-	Findings are non-trivial and significance.
-	The paper's is clear and easy to read.

Weaknesses:
-	Technical aspects are inadequately addressed.
-	Deeper analysis of the real-world dataset (APW-Wild) is absent from the paper.

The paper adeptly identifies a critical gap in prior research by underscoring the lack of user studies proving that perturbations are subtle enough to avoid causing discomfort to end-users. By delving into this often-neglected practical aspect, the paper contributes to the understanding of adversarial attacks. The qualitative study's findings, particularly those pertaining to research questions 2 and 4, provide valuable insights into users' ability to identify phishing websites amid perturbative adversarial techniques.

Nevertheless, certain enhancements could elevate the study's rigor. Firstly, a more in-depth quantitative analysis would be beneficial from a technical perspective. While the primary focus is on user studies, considering other variants of adversarial phishing methods beyond perturbation types may enrich the study. For instance, comparing multiple perturbation techniques within the same category could offer a compelling evaluation. Secondly, in the analysis related to Research Question 2 which have been conducted on the APW dataset, although the authors acknowledged inspecting the APW-Wild dataset and identifying blurry logos and background patterns, a clearer alignment with synthetic datasets (APW-LAB) is recommended. In this work, APW-Wild is employed as a single variable in the regression, but further clarification on the types of perturbation in APW-Wild would enhance the paper's methodological transparency.

**Questions:**

- could you provide more detailed quantitative analysis results?

- could you provide more detailed information and analysis results on APW-wild case?

**Reviewer Confidence:**

3: The reviewer is confident but not certain that the evaluation is correct

**Scope:**

3: The work is somewhat relevant to the Web and to the track, and is of narrow interest to a sub-community

---

### Official Review · Reviewer_qdhU · 2023-11-23

**Novelty:** 5
**Technical Quality:** 5

**Review:**

This paper addresses the vulnerability of ML-PWD to adversarial attacks, highlighting the need to examine their impact on end users. The study involves two user surveys evaluating both synthetically crafted adversarial phishing webpages and real-world examples. The paper concludes that adversarial phishing poses a threat to both users and ML-PWD, however, not all adversarial perturbations are equally effective.
That is a much needed research, the paper introduces a novel perspective of the problem by focusing on user perception. The paper has done a good job on performing a user study and the results are interesting.


That being said, the work has limitations and the limitations are neither explicitly highlighted nor discussed. The study is just a proof of concept to perform user study and the results cannot be used to make a generic conclusion about the effectiveness of adversarial pages.
It is important to acknowledge that there are diverse methods for perturbing phishing pages, and adversaries may target different features in various ways, beyond the scope of this study. It is crucial to underscore these limitations, align the conclusions and paper title with the stated claims, and clarify that the results may not be generalized to other scenarios.


Moreover, the paper lacks a comprehensive table or description detailing the types of features used in the study, which would provide insights into the variety of possible evasions.

**Questions:**

-

**Ethics Review Description:**

-

**Reviewer Confidence:**

3: The reviewer is confident but not certain that the evaluation is correct

**Scope:**

4: The work is relevant to the Web and to the track, and is of broad interest to the community

---

### Official Review · Reviewer_12Cy · 2023-11-24

**Novelty:** 5
**Technical Quality:** 5

**Review:**

The paper focuses on users’ perception on assessing adversarial phishing web sites by checking if users are vulnerable to perturbed cases. The paper conducted two studies with 470 users. The paper concludes that adversarial phishing is equally risky on users and AI-driven detection systems. The paper also concludes that not all the perturbation mechanism are successful on users compared to AI machines.



**Strength**
The paper focuses on an interesting and less well-studied area.
The number of participants are large and diverse.

**weaknesses**
The perturbation mechanism needs to be justified more clearly.


I would like to thank the authors for defining a project in this space. Understanding the weaknesses and flaws in human perception in phishing has never been easy and the adversarial phishing could make the problem space even more complex.

The paper is well-structured. It contains almost all the necessary details to understand the work and follow the steps. I liked the questions that were asked in the paper. They are generally well-justified and in the scope of the project.

The finding was also interesting. The fact that the paper empirically showed that not all the perturbation approaches are equally successful is interesting. The  findings on the impact of perturbation mechanisms on participants was reasonable. That said, there are a few areas that can enhance the quality of the paper even further.

It would be helpful to provide more detail on how the perturbation mechanism is being designed. The current version seems a bit random. Was the perturbation mechanism defined a systematic way? It would be very helpful to provide more details on this step for the followup work in this domain and  to evaluate the generalizability of the findings.

All in all, I found the paper interesting with a reasonable scientific pipeline and unique insights.

**Questions:**

Please elaborate more on the perturbation method and if/how the findings can be generalized?
Also, the ethics section needs to cover more detail. Section 4.3 doesn't really discuss potential risks. were the phishing pages accessible to users not in the participant group?

**Ethics Review Description:**

While it seems the paper has followed some ethics rules, but the paper needs to discuss that section in a greater detail. Where the phishing pages accessible to users other than the target participants?

**Ethics Review Flag:**

Yes

**Reviewer Confidence:**

4: The reviewer is certain that the evaluation is correct and very familiar with the relevant literature

**Scope:**

4: The work is relevant to the Web and to the track, and is of broad interest to the community

---

### Decision · Program_Chairs · 2024-01-22

**Decision:**

Accept (Oral)

**Comment:**

# Summary

 Prior research has shown that attacks can (and do) perturb their phishing websites to bypass detection, and research has also shown how this can be done using automated techniques such as adversarial ML. This paper asks the question: "Can humans detect these adversarial phishing websites compared to normal phishing websites?" This is an important question to answer, as it gets to the core of the benefit (or lack therefor) of adversarial phishing websites. The paper presents the design of a user study to understand human perception of adversarial phishing websites.

 # Strengths

 + Interesting research area.
 + Large study (470 participants).
 + Paper brings a novel perspective to the problem of adversarial phishing.

 # Weaknesses

 - Paper lacks details re: perturbation mechanisms (will be addressed).
 - Paper lacks analysis of dataset (will be addressed).

 # Recommendation

 Overall, the PC was excited that this paper explores the question of how adversarial phishing websites interact with humans. This is likely to be the first of many steps in this research direction, as the reviewers also have plenty of suggestions on future research directions for this line of work. In addition, the changes proposed by the authors (based on the reviewers' feedback) will significantly strengthen the paper, and the recommendation for Strong Accept is based on all the changes proposed by the authors being made.

 ---